# Light-Induced Protein Clustering for Optogenetic Interference and Protein Interaction Analysis in *Drosophila* S2 Cells

**DOI:** 10.3390/biom9020061

**Published:** 2019-02-12

**Authors:** Mariana Osswald, A. Filipa Santos, Eurico Morais-de-Sá

**Affiliations:** 1Epithelial Polarity and Cell Division, i3S, Instituto de Investigação e Inovação em Saúde, Universidade do Porto, 4200-135 Porto, Portugal; mariana.osswald@ibmc.up.pt (M.O.); afsantos@i3s.up.pt (A.F.S.); 2IBMC, Instituto de Biologia Molecular e Celular, Universidade do Porto, 4200-135 Porto, Portugal

**Keywords:** *Drosophila*, Schneider 2 cells, optogenetics, LARIAT, mitosis, cell polarity, Mps1, aPKC

## Abstract

*Drosophila* Schneider 2 (S2) cells are a simple and powerful system commonly used in cell biology because they are well suited for high resolution microscopy and RNAi-mediated depletion. However, understanding dynamic processes, such as cell division, also requires methodology to interfere with protein function with high spatiotemporal control. In this research study, we report the adaptation of an optogenetic tool to *Drosophila* S2 cells. Light-activated reversible inhibition by assembled trap (LARIAT) relies on the rapid light-dependent heterodimerization between cryptochrome 2 (CRY2) and cryptochrome-interacting bHLH 1 (CIB1) to form large protein clusters. An anti-green fluorescent protein (GFP) nanobody fused with CRY2 allows this method to quickly trap any GFP-tagged protein in these light-induced protein clusters. We evaluated clustering kinetics in response to light for different LARIAT modules, and showed the ability of GFP-LARIAT to inactivate the mitotic protein Mps1 and to disrupt the membrane localization of the polarity regulator Lethal Giant Larvae (Lgl). Moreover, we validated light-induced co-clustering assays to assess protein-protein interactions in S2 cells. In conclusion, GFP-based LARIAT is a versatile tool to answer different biological questions, since it enables probing of dynamic processes and protein-protein interactions with high spatiotemporal resolution in *Drosophila* S2 cells.

## 1. Introduction

Understanding the mechanisms that control the dynamic behavior of most cellular processes requires an experimental setup that enables detailed visualization with high-resolution microscopy and that is amenable to genetic manipulation. *Drosophila* Schneider 2 (S2) cells have long been recognized as a powerful cell culture model to study the underlying mechanisms controlling cell division and are particularly well suited for high-throughput RNA interference screens via double-stranded RNAs [1,2,3,4,5]. Moreover, S2 cells provide a reduced system for the molecular dissection at the cell autonomous level of processes that require reorganization of the cytoskeleton and the plasma membrane in a particular axis, such as cell motility, cell polarity, and oriented cell division [6,7,8]. Importantly, investigation of these very dynamic cellular processes requires progression from established genetic approaches to methodologies that perturb protein function with high spatial and temporal control. Temporal control can be achieved through chemical inhibition, but this lacks spatial resolution, reversibility, and shows common off-target effects. Thus, the advances in optogenetic tools that enable rapid modulation of protein activity with light provide unprecedented spatiotemporal control over dynamic cellular processes [9,10] and are likely to bring fruitful times for cell biologists.

Light-activated reversible inhibition by assembled trap (LARIAT) was developed in mammalian cells to manipulate protein function through light-inducible and reversible formation of multimeric protein clusters [11]. This tool combines the photoreceptor ryptochrome 2 (CRY2) with cryptochrome-interacting bHLH 1 (CIB1) oligomers. CRY2 forms both homo-oligomers and heterodimers with CIB1 within seconds of blue-light exposure [12]. This was coupled with a fusion between CIB1 and the multimerization domain (MP) of Ca^2+^/Calmodulin-dependent protein kinase II α (CaMKII) to drive the formation of large clusters (Figure 1). In addition, CRY2 fused with an anti-green fluorescent protein (GFP) nanobody sequesters GFP-tagged proteins in the light-induced clusters in a reversible manner [11]. LARIAT is, therefore, a versatile tool that has been exploited in mammalian cells to disrupt a variety of pathways, including Rho GTPase signaling, the microtubule cytoskeleton, and membrane trafficking [11,13], as well as cell adhesion and actomyosin contractility in *Drosophila* tissues [14,15]. However, these approaches have yet to be implemented in *Drosophila* cell culture models.

In this study, we adapted optogenetic clustering to *Drosophila* S2 cells, which generates an inducible module for expression of LARIAT components. To validate LARIAT as a tool to study cell division in S2 cells, we provide an example of the application showing that LARIAT can be used to trap and inactivate the key regulator of mitotic fidelity monopolar spindle 1 (Mps1). Moreover, we evaluated the potential of LARIAT in S2 cells for the molecular dissection of other processes associated with cell division, such as cortical cell polarity. Both asymmetric stem cell division [16,17] and mitotic spindle orientation in some epithelial tissues [8,18,19,20] rely on the dynamic control of two conserved regulators of cortical polarity: the atypical protein kinase C (aPKC) complex and Lethal Giant Larvae (Lgl). Lethal Giant Larvae cortical localization is reproduced in *Drosophila* S2 cells, which have previously been used to dissect the molecular mechanisms regulating Lgl subcellular localization [8,16,21,22]. We, thus, monitored the ability of LARIAT to delocalize the membrane-associated protein Lgl and to determine protein interactions within the aPKC complex in living cells. Hence, this new tool expands the power of *Drosophila* S2 cells as a model for spatiotemporal investigation of mechanisms controlling cell division and cell polarity, which are two interconnected processes whose proper understanding demands the ability to interfere with protein function and to evaluate protein interactions with high temporal control.

## 2. Materials and Methods

### 2.1. Molecular Biology

We cloned all LARIAT modules (Figure 2), aPKC, aPKCΔN, and partitioning defective protein 6 (Par6) into *Drosophila* Gateway vectors. We started by inserting them into pENTR through FastCloning [23] with the respective primers (indicated in Table 1). We amplified the CIB1 N-terminal (CIBN, amino acids 1–170) fused with mCerulean and the oligomerization domain from CaMKIIα (amino acids 315–478) from pCMV-CIB1-mCerulean-CaMKIIα multimerization domain from (MP) (Addgene plasmid #58366, gift from Won do Heo, Institute for Basic Science (IBS) and Korea Advanced Institute of Science and Technology (KAIST), Daejeon, Republic of Korea) [11], CRY2 photolyase homology region (amino acids 1–498) fused with an anti-GFP nanobody (V_H_H) from pCMV-SNAP-CRY2-V_H_H(GFP) (Addgene plasmid #58370, gift from Won do Heo) [11] and aPKC, aPKCΔN (amino acids 180–606), and Par6 from their respective complementary DNAs (Berkeley *Drosophila* Genome Project (BDGP) Gold collection). After PCR amplification with Phusion Polymerase (New England Biolabs (NEB), Ipswich, MA, USA), we destroyed template DNA with DpnI (NEB) and transformed chemically competent TOP10 *Escherichia coli* cells with the insert and pENTR PCR mixes. To generate pENTR-CIBN-MP, we removed mCerulean from pENTR-CIBN-mCerulean-MP through restriction with AgeI (NEB) and BspEI (NEB). pENTR-CRY2olig(E490G)-V_H_H was generated through site-directed mutagenesis as in FastCloning [23] (Table 1). Then, we recombined aPKC and aPKCΔN into pHGW, Par6 into pHWR, CIBN constructs into pHW, and CRY2 constructs into pHRW through LR Clonase II (Thermo Fisher Scientific, Waltham, MA, USA)-mediated recombination.

For S2 cell transfection, we joined CIBN and CRY2 constructs in the same plasmid through Circular Polymerase Extension Cloning (CPEC) [24]. To this end, we created fragments with overlapping extremities by amplification with Gateway primers (Table 1) of RFP-tagged CRY2 constructs, including promoter and terminator regions, and both pHW-CIBN plasmids. CIBN and CRY2 fragments were combined through overlap annealing and Phusion DNA polymerase (NEB)-mediated extension. To create a non-fluorescent LARIAT plasmid, we recombined CRY2-V_H_H into pHFW harboring an additional AfeI restriction site before the Hsp70 promoter. This site had been previously created through site-directed mutagenesis. Then, the CRY2-V_H_H expression cassette was excised with AfeI (NEB) and PvuI (NEB) and inserted into pHW-CIBN-MP previously linearized with PmeI (NEB) and PvuI (NEB). All expression plasmids were sequenced prior to *Drosophila* S2 cell transfection and will be made available through Addgene (Watertown, MA, USA). pHWG-Lgl [8], mCherry-α-Tubulin [2], and pMT-EGFP-Mps1 and pMT-EGFP [25] have been described previously.

### 2.2. Schneider 2 Cell Culture, Transfection, and Imaging

*Drosophila* S2-DGRC cells (*Drosophila* Genomics Resource Center) were cultured at 25 °C in Schneider’s Insect medium (Sigma-Aldrich, St. Louis, MO, USA) supplemented with 10% fetal bovine serum. We used the Effectene Transfection Reagent (QIAGEN, Hilden, Germany), according to the manufacturer’s instruction to transiently transfect S2 cells with the plasmids indicated for the experiments in each figure. After transfection, cells were incubated in the dark, but no measure was taken to limit light exposure during cell manipulation prior to LARIAT expression. We induced the expression of all Hsp70 promoter-controlled constructs by incubating the cells at 37 °C for 45 min at least 6 h prior to imaging and the GFP-Mps1 or GFP construct with 100 μM CuSO_4_ at least 12 h prior to imaging. After LARIAT expression was induced, we kept cells in the dark and used a 593 nm LED light (Super Bright LEDs, St. Louis, MO, USA) during manipulation when necessary. For imaging, we seeded S2 cells on glass-bottom dishes (MatTek Corporation, Ashland, MA, USA) coated with Concanavalin A (for experiments comparing LARIAT constructs, assessing mitotic progression or involving Mps1 clustering) or poly-l-lysine (for Lgl and aPKC clustering experiments). Cells were imaged between day 2 and day 7 post-transfection with an Andor XD Revolution Spinning Disk Confocal system (Andor Technologies, Belfast, United Kingdom) equipped with two solid state lasers—488 nm and 561 nm—an iXonEM+ DU-897 EMCCD camera and a Yokogawa CSU-22 unit built on an inverted Olympus IX81 microscope (Olympus Corporation, Tokyo, Japan) with a PLAPON 60x/NA 1.42 or a UPLSAPO 100x/NA 1.40 objective. Cells for mitotic timing experiments were imaged in a Nikon TE2000 microscope (Nikon, Tokyo, Japan) equipped with a modified Yokogawa CSU-X1 unit (Solamere Technology, Salt Lake City, UT, USA), an FW-1000 filter-wheel (ASI), and an iXon+ DU897 EM-CCD (Andor Technologies, Belfast, UK) and controlled by NIS-Elements via a DAC board (PCI-6733, National Instruments, Austin, TX, USA). An acousto-optic tunable filter (Gooch&Housego, Ilminster, United Kingdom, model R64040-150) controlled fluorophore excitation by two sapphire lasers −488 nm and 561 nm. We used an oil-immersion 60x 1.4 NA Plan-Apo DIC CFI (Nikon, VC series). All live-imaging experiments were performed at a controlled temperature of 25 °C.

### 2.3. Immunofluorescence

The Flag tagged CRY2 was validated through immunofluorescence. *Drosophila* S2 cells were transiently transfected with pMT-EGFP and pH-CIBN-MP-HF-CRY2-V_H_H(GFP) with the Effectene Transfection Reagent (QIAGEN, Hilden, Germany). To induce clustering, cells were exposed to a 472 nm LED light (Super Bright LEDs) prior to fixation. 10^5^ cells were then centrifuged onto slides for 5 min at 1000 rpm (Shandon Cytospin 2, Thermo Fisher Scientific), fixed with 3.7% formaldehyde (Sigma-Aldrich) in PHEM (60 mM PIPES, 25 mM HEPES, pH 7.0, 10 mM EGTA, 4 mM Mg_2_SO_4_) for 12 min, and permeabilized with 0.5% Triton X-100 (Sigma-Aldrich) in phosphate-buffered saline (PBS) three times for 5 min each. Fixed cells were blocked for 1 h with 10% fetal bovine serum (FBS) diluted in PBS-0.05% Tween-20 (Sigma-Aldrich) and afterwards incubated overnight at 4 °C with mouse anti-Flag (1:250, M2 clone, Sigma-Aldrich) diluted in the blocking solution. Subsequently, cells were washed three times with PBS-0.05% Tween20 and incubated for 1 h at room temperature with anti-mouse Alexa 647 (1:1000, Invitrogen, Carlsbad, CA, USA) diluted in the blocking solution. After washing three times with PBS-0.05% Tween20, slides were mounted with Vectashield Mounting Medium with DAPI (Vector Laboratories, Burlingame, CA, USA). Slides were imaged with a LEICA TCS SP5 II laser scanning confocal microscope (Leica Microsystems, Wetzlar, Germany).

### 2.4. Image Analysis

Image sequences were assembled and analyzed using Fiji/ImageJ [26]. To quantify protein clustering, we calculated the coefficient of variation in the cytoplasm (region of interest (ROI) over 40 μm^2^ in size). We measured the standard deviation of fluorescence intensity in the ROI and divided it by the mean fluorescence intensity in the same region corrected for the background signal. Then, we normalized the coefficient of variation to its mean value before cluster formation and plotted it through time with GraphPad Prism 8 (GraphPad Software, Inc., La Jolla; CA, USA). To determine how long it takes to revert to 50% and 10% of maximal clustering in Figure A1, clustering curves were smoothed with a moving average. For the experiments addressing the effect of clusters on mitotic progression and Mps1 clustering, mitotic timing was determined as the time between nuclear envelope breakdown (NEB, defined as the first frame where tubulin signal is present inside the nuclear area) and anaphase onset (AO, defined as the first frame where movement toward opposite sides by both spindle halves can be seen). To determine which cells expressed LARIAT constructs, we measured clustering in the first three time points after clustering stimulation through GFP or GFP-Mps1 coefficient of variation. We measured GFP-Mps1, Lgl-GFP, and RFP-CRY2 relative expression levels by extracting the average fluorescence intensity in the cell and correcting for the background signal. Fluorescence intensity profiles along the lines were obtained with the Plot Profile tool from Fiji. Kymographs were made by stacking the indicated line ROIs from every timepoint sequentially. The statistical analysis was completed with GraphPad Prism 8.

## 3. Results and Discussion

### 3.1. Generation and Characterization of Heat Shock-Inducible LARIAT Modules in S2 Cells

To build a toolbox for interference with protein function through optogenetic clustering in *Drosophila* S2 cells, we adapted LARIAT from a set of constructs previously developed for human cells [11]. We cloned CIBN fused with a multimeric protein (MP, the oligomerization domain from CaMKIIα) and CRY2 photolyase homology region (PHR) into a single plasmid to ensure constant relative amounts of CIBN and CRY2 in transfected cells (Figure 2 and Figure 3). Fusion of CRY2 with an anti-GFP nanobody (V_H_H) allows clustering of GFP-tagged proteins without additional cloning. Lastly, we used an inducible Hsp70 promoter to allow temporal control over LARIAT expression, which obviates the need to limit light exposure during the cell culture.

In *Drosophila* S2 cells, CRY2 clustered with CIBN-MP in under 30 s after a single pulse stimulation with a 488 nm laser (construct 3 in Figure 3A, Appendix A), a similar timeframe to the one for human cells [11]. The cluster size increased for about 5 min upon the pulse of light and then disassembled spontaneously within around 30 min with no protein degradation (Figure 3A,C). Clustering sequesters RFP-CRY2 in puncta and disrupts fluorescence homogeneity in the cytoplasm. Thus, we evaluated clustering kinetics by measuring fluorescence heterogeneity with the coefficient of variation of RFP-CRY2 fluorescence (standard deviation normalized to average intensity). This allowed us to quantitatively compare the kinetics of cluster assembly and reversibility in different imaging conditions and for different optogenetic modules (Figure 3B). The reversion half-life is longer in S2 cells (t_1/2_ = 17.5, Figure A1C) when compared with human cells, where LARIAT-induced clusters return to diffusive state within 10 min in the dark [11]. Modulation of laser intensity controls the extent of CRY2/CIBN clustering in S2 cells (Figure A1A), as previously observed in mammalian cells [11]. However, we observed longer reversion periods in S2 cells even when using very low laser intensity (t_1/2_ = 16, Figure A1D), for which protein clusters are barely detected. Since S2 cell experiments are performed at a lower temperature (25 °C), the difference in cluster half-life may be related to an effect of temperature in the rate of CRY2-CIBN complex formation or disassembly.

The clustering efficiency of different CRY2 variants has also been correlated with their expression levels [27,28]. Accordingly, we observed faster kinetics of LARIAT-mediated clustering for cells expressing higher levels of RFP-CRY2/CIBN (Figure A1B). Thus, to enable a comparison between the different LARIAT modules (constructs 2, 3, 5 in Figure 3A), we restricted the analysis to cells within a range of equivalent expression of LARIAT components measured by RFP fluorescence (Figure 3C). The presence of the mCerulean tag in CIBN decreased clustering for the same expression levels and light stimulation conditions (Figure 3A,B), which suggests that mCerulean could be partially blocking CRY2 binding to CIBN. Given that mCerulean is a β barrel with its N-terminus and C-terminus sticking out from the same side [29], we speculate that fusion with mCerulean positions CIBN towards the inside of the oligomer, and partially obstructs access of CRY2 to CIBN.

In mammalian cells, CRY2 oligomerization by itself was shown to cluster target proteins [30]. However, when expressed by themselves in our assay, we did not detect clustering with either CRY2 or CRY2olig, which is a CRY2 variant with enhanced oligomerization ability due to the point mutation E490G [27,31] (Figure 3). This could result from lower expression levels of CRY2 variants driven by the inducible heat-shock promoter relatively to the strong constitutive expression induced in mammalian cells by the CMV promoter. When co-expressed with CIBN-MP in S2 cells, CRY2olig did not cluster more efficiently than wild-type CRY2 and the clusters disassembled faster (Figure 3), which indicates that this LARIAT module should be considered for experiments that require faster reversibility of protein interference.

Altogether, these experiments highlight the potential of adjusting expression levels and laser intensity to control clustering kinetics and suggest that LARIAT clustering can be fine-tuned by modulating CRY2 propensity to oligomerize. It would be useful to test the effect of CRY2clust [28], CRY2low [31], and other variants that affect photocycle kinetics [32] on LARIAT since they will enable further modulation of cluster assembly and stability.

### 3.2. Interference with GFP-Tagged Proteins by Optogenetic Clustering

The presence of LARIAT clusters does not affect mammalian cell viability [11]. We, therefore, proceeded to evaluate the potential of GFP-based LARIAT to dissect mitotic processes. We started by testing if the presence of protein clusters by themselves could interfere with mitotic progression and spindle organization. With that purpose, we generated LARIAT constructs where the RFP-tag for CRY2 was replaced by a FLAG tag to enable proper visualization of microtubule organization using mCherry-tubulin, while co-expression of GFP was used to validate cluster formation (Figure A2). Formation of light-induced protein clusters enables the assembly of normal bipolar spindles and does not interfere with the time from the nuclear envelope breakdown to anaphase onset (Figure 4). This suggests that the formation of CRY2-CIBN clusters does not affect mitotic spindle assembly or chromosome congression at the metaphase plate since any defects on these processes would introduce significant delays in the mitotic exit.

As a proof of concept that LARIAT can be used to dissect cell division in *Drosophila* S2 cells, we then tested the ability of GFP-based LARIAT to sequester and inactivate the mitotic kinase Mps1. Mps1 is a conserved upstream regulator of the Spindle Assembly Checkpoint (SAC), which is a molecular mechanism that ensures chromosome segregation fidelity during mitosis. This prevents chromatid separation until all chromosomes are properly attached to opposite spindle poles [33,34]. Upon mitotic entry, Mps1 accumulates at the kinetochores to promote SAC signaling [35,36]. When co-expressed with LARIAT, Mps1-GFP rapidly clustered during mitosis upon blue light exposure (Figure 5A). The Mps1-GFP signal decreased in the cytoplasm and increased at the RFP-CRY2-V_H_H puncta, which indicates that Mps1-GFP was efficiently trapped in the optogenetic clusters (Figure 5B). As cells reach metaphase, Mps1 must be removed from kinetochores and inactivated to enable anaphase onset [37,38,39,40]. Accordingly, high overexpression levels of Mps1 leads to persistent SAC signaling that blocks cells in mitosis during a prolonged period of time [25] (Figure 5C,D). Light-induced clustering of Mps1-GFP is sufficient to revert this phenotype (Figure 5C,D), which indicates that optogenetic trapping of Mps1 leads to inactivation.

We also determined the ability of GFP-based LARIAT to disrupt the localization of proteins associated with the plasma membrane and the underlying cortex, and which could play a role in processes associated with cell division. To this end, we analyzed Lgl, whose subcellular distribution controls the segregation of cell fate determinants during asymmetric cell division [16,17,41,42] and is regulated to enable proper mitotic spindle orientation in *Drosophila* epithelial tissues [8,20]. Lethal Giant Larvae localizes at the cell cortex mainly through association with plasma membrane phosphoinositides [43,44]. Moreover, it also interacts with the actomyosin cytoskeleton [16,45]. Prior to clustering stimulation, RFP-CRY2-V_H_H accumulated at the cell cortex, which indicates that the GFP-nanobody recruited CRY2 to the same subcellular regions as Lgl-GFP (Figure 6A). Blue light exposure led to immediate cluster formation, which ultimately resulted in Lgl-GFP removal from the cortex (Figure 6A,A’ and Appendix A). However, light-induced disruption of Lgl localization failed for a significant number of cells. We found that efficient removal of Lgl from the cortex requires high levels of RFP-CRY2 relative to Lgl-GFP (Figure 6B,B’,C), which highlights the importance of expressing LARIAT modules in excess of the target protein. This can be ensured by keeping target proteins at endogenous levels, for example, using GFP knock-in lines. These lines can be easily generated through CRISPR/Cas9 in *Drosophila* S2 cells [46].

### 3.3. Analysis of Protein-Protein Interactions via Optogenetic Co-Clustering

CRY2olig has previously been applied as a tool to measure protein interactions by evaluating recruitment of binding partners to light-induced clusters of target proteins [27]. This system requires the production of constructs to tag any new “bait” protein with CRY2olig. We evaluated if we could use GFP-based LARIAT to probe interactions between GFP tagged proteins, for which there are many constructs available, and putative binding partners co-expressed with a different fluorescent tag. As a proof of concept, we evaluated if this system could detect the formation of the aPKC/Par-6 complex, which regulates cortical polarity during asymmetric cell division [16,42]. aPKC-Par-6 interaction is mediated by heterodimerization between the N-terminal PB1 (Phox and Bem1) domains of aPKC and Par6 [47,48,49]. Live imaging shows that, upon blue-light stimulation, Par6-RFP redistributed rapidly and co-localized with the GFP-aPKC clusters, which confirms that the two proteins interact with high-affinity (Figure 7A,B). The LARIAT system can be used for structure-function analysis since removing the N-terminal region with the PB1 domain of aPKC (GFP-aPKCΔN) was sufficient to fully abolish the localization of Par6-RFP in the optogenetic clusters (Figure 7C,D). Thus, this is a particularly useful approach in *Drosophila* S2 cells, which can be used both with live and fixed cells to complement well-established biochemical methodology to identify protein interactions. Furthermore, the ability to perform live examination of interactions at the single cell level enables precise analysis during specific periods of the cell cycle and can be used to dissect the effect of pharmacological manipulation of signaling pathways and cytoskeleton organization.

## 4. Conclusions

In this work, we generated optogenetic modules to study protein function and interactions in *Drosophila* S2 cells. *Drosophila* LARIAT allows light-controlled clustering of proteins at the single cell level, which can be used to interfere with protein localization and functioning in S2 cells. This is particularly useful for proteins involved in multiple processes, whose depletion leads to pleiotropic phenotypes. Furthermore, we also expand the application of LARIAT to probe for protein interactions. Since this is accomplished at the single cell level through live imaging, it can be used to test protein interactions at specific phases of the cell cycle. Lastly, comparison of different LARIAT variants provides relevant information to guide the generation of new optogenetic modules, and our data suggests that tweaking CRY2 oligomerization properties may produce LARIAT modules with faster reversibility. This would provide even tighter temporal control over the protein function.

## Figures and Tables

**Figure 1 biomolecules-09-00061-f001:**
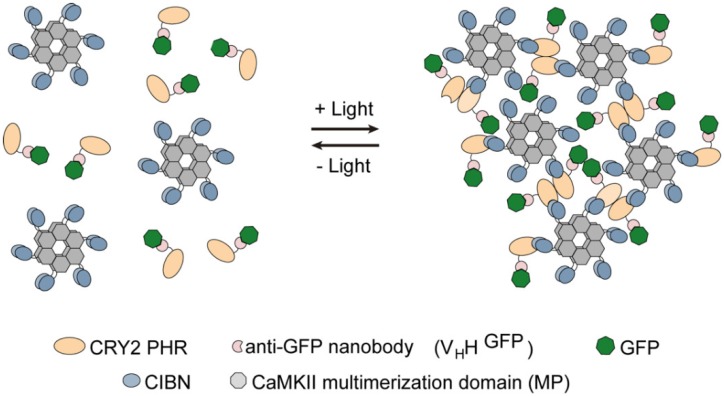
Schematic representation of light-activated reversible inhibition by assembled trap (LARIAT)-mediated optogenetic clustering. It enables optogenetic clustering of target proteins to interfere with their function and to probe interactions. Cryptochrome-interacting bHLH N-terminal (CIBN) fused with the multimerization domain from CaMKIIα (MP) forms dodecamers in the cytoplasm. The cryptochrome 2 (CRY2) photolyase homology region (PHR) is fused with an anti-GFP nanobody that binds specifically to GFP-tagged proteins. Blue light triggers CRY2 oligomerization and binding to CIBN and consequently the formation of clusters to trap GFP-tagged proteins. In the dark, CRY2 reverts spontaneously to its ground state and the clusters disassemble.

**Figure 2 biomolecules-09-00061-f002:**
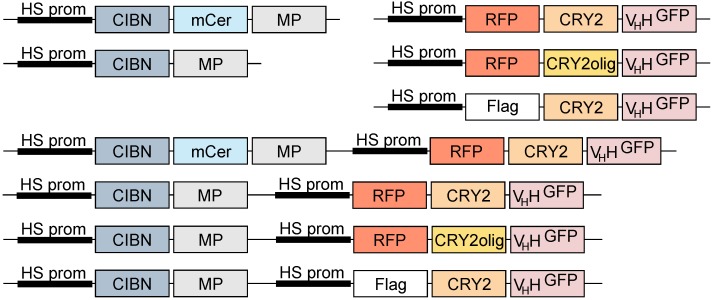
Schematic representation of LARIAT plasmids for *Drosophila* Schneider 2 (S2) cells. LARIAT modules were cloned downstream of the Hsp70 promoter. Untagged and mCerulean-tagged CIBN are fused with the MP. Flag- or red fluorescence protein (RFP-) tagged CRY2 photolyase homology region are fused with an anti-GFP nanobody (V_H_H^GFP^). CRY2olig oligomerization ability is enhanced due to a single point mutation—E490G. Different CRY2 and CIBN expression cassettes were combined in the same vectors for efficient co-expression.

**Figure 3 biomolecules-09-00061-f003:**
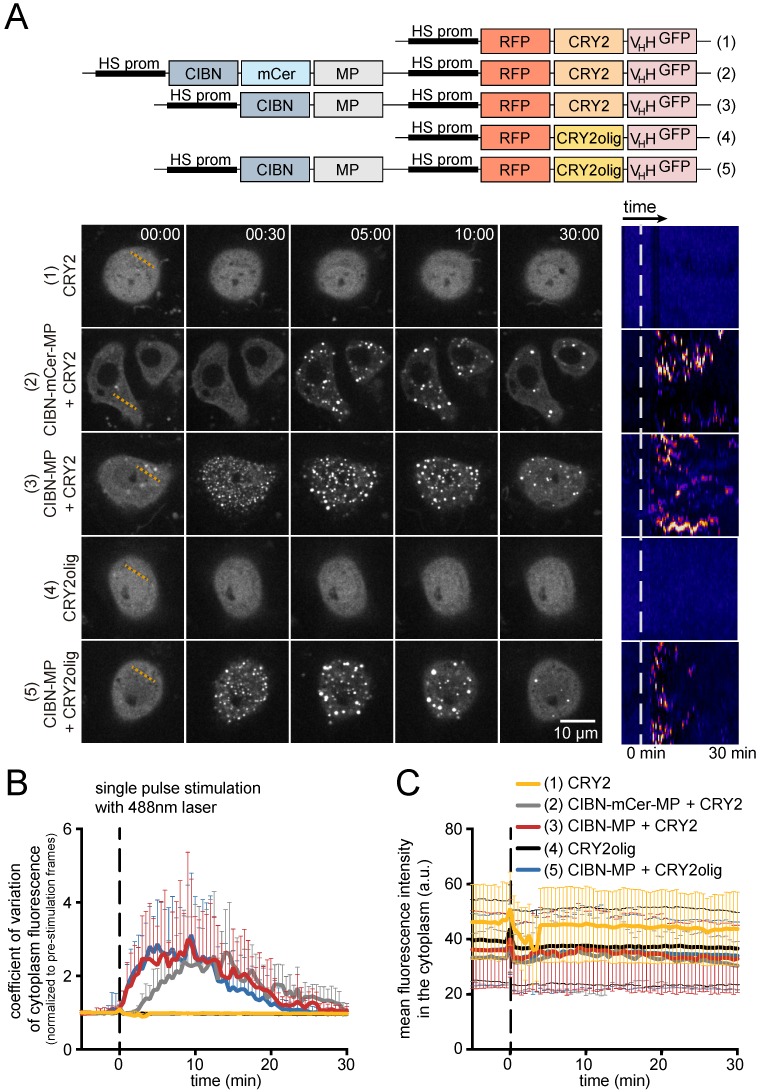
A *Drosophila* S2 cell LARIAT toolbox. (**A**) Representative timepoints and Z-sections of *Drosophila* S2 cells expressing the indicated LARIAT modules. Transfected cells were kept in the dark prior to live-imaging. Pseudo-colored kymographs made along the orange dashed lines represent cluster formation and disassembly after stimulation with a 488 nm laser set to 1 mW laser power for 300 msec at minute 0 (white dashed line). (**B**) Clustering was quantified with the coefficient of variation of cytoplasm fluorescence and (**C**) mean fluorescence intensity was plotted along time. Graphs display mean ± standard deviation (SD). *n*_(CRY2)_ = 9 cells, *n*_(CIBN-mCer-MP+CRY2)_ = 5 cells, *n*_(CIBN-MP+CRY2)_ = 15 cells, *n*_(CRY2olig)_ = 10 cells, *n*_(CIBN-MP+CRY2olig)_ = 13 cells.

**Figure 4 biomolecules-09-00061-f004:**
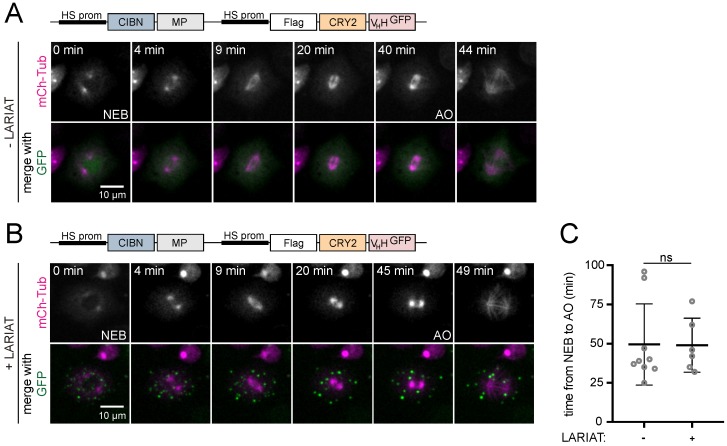
LARIAT clusters do not affect bipolar spindle assembly and mitotic progression. Clustering effect on mitotic progression was analyzed through live-imaging of *Drosophila* S2 cells expressing GFP, mCherry-α-Tubulin (**A**) alone, and co-transfected with CIBN-MP and Flag-CRY2-V_H_H (**B**). The 488 nm laser used to image GFP triggered clustering since the prophase stage. LARIAT expression was confirmed by measuring GFP coefficient of variation in the first three frames of each movie. (**C**) Graph shows average mitotic timing ± SD. Mitotic timing is defined as the time it takes for a cell to progress from nuclear envelope breakdown (NEB) to anaphase onset (AO). Not significant (ns) *p* = 0.2149 (Mann-Whitney test).

**Figure 5 biomolecules-09-00061-f005:**
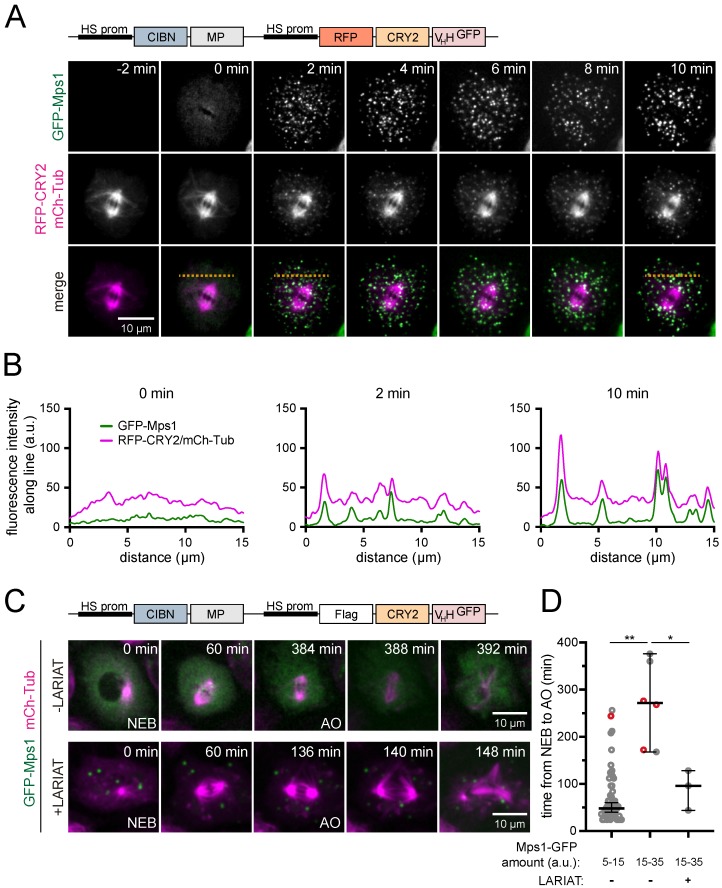
GFP-Mps1 is efficiently trapped in LARIAT clusters. Clustering was analyzed through live-imaging of *Drosophila* S2 cells expressing GFP-Mps1, mCherry-α-Tubulin, CIBN-MP, and RFP-CRY2-V_H_H. The 488 nm laser used to image GFP-Mps1 from timepoint 0 onwards triggered clustering. (**A**) Representative stills and (**B**) intensity profiles for GFP-Mps1 and RFP-CRY2-V_H_H/mCherry-α-Tubulin along the orange dashed line are shown. (**C**) Representative stills of GFP-Mps1, mCherry-α-Tubulin, CIBN-MP, and Flag-CRY2-V_H_H transfected cells analyzed through live imaging during cell division. (**D**) Graph shows median mitotic timing ± 95% confidence interval. Mitotic timing is defined as the time it takes for a cell to progress from nuclear envelope breakdown (NEB) to anaphase onset (AO). GFP-Mps1 levels were determined by measuring GFP fluorescence in the cell cytoplasm in the first frame and LARIAT clustering was confirmed by measuring the GFP-Mps1 coefficient of variation in the first three frames of each movie. Red circles in the graph represent cells that did not exit mitosis during the time they were imaged. ** *p* < 0.001, * *p* < 0.05 (Mann-Whitney test).

**Figure 6 biomolecules-09-00061-f006:**
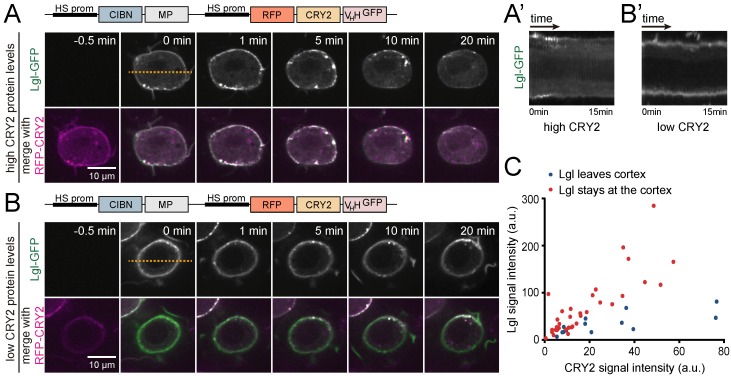
High levels of LARIAT delocalize membrane-associated Lgl-GFP. (**A**,**B**) Representative single z-sections and (**A’**,**B’**) respective kymographs of *Drosophila* S2 cells transiently transfected with Lgl-GFP, CIBN-MP, and RFP-CRY2-V_H_H expressing (**A**,**A’**) higher or (**B**,**B’**) lower levels of CRY2 relative to Lgl. (**C**) Dispersion plot for average Lgl-GFP and RFP-CRY2-V_H_H signal intensity in cells in which Lgl left (blue) or remained in (red) the cortex after clustering. Each dot represents a single cell.

**Figure 7 biomolecules-09-00061-f007:**
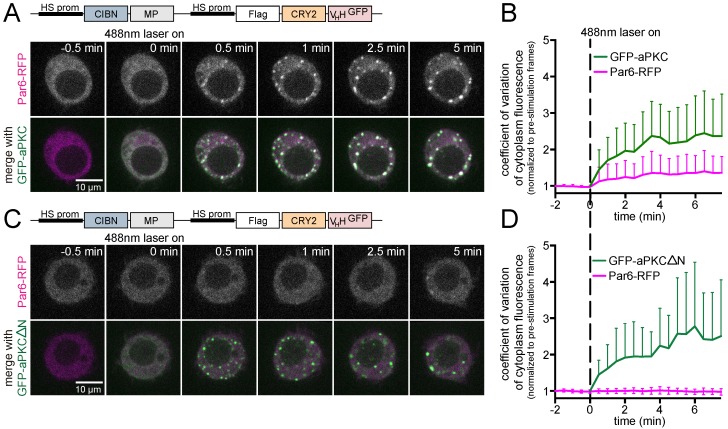
LARIAT detects aPKC and Par6 interaction in single cells. Representative timepoints of *Drosophila* S2 cells expressing Par6-RFP, CIBN-MP, Flag-CRY2-V_H_H, and (**A**) GFP-aPKC or (**C**) GFP-aPKCΔN analyzed through live-imaging. (**B**,**D**) Par6-RFP and (**B**) GFP-aPKC or (**D**) GFP-aPKCΔN clustering is represented by the coefficient of variation of cytoplasm fluorescence. Graphs display mean ± SD. *n* ≥ 15 cells for each condition.

**Table 1 biomolecules-09-00061-t001:** List of primers for molecular cloning.

Purpose	Primer Name	Primer Sequence (5′–3′)
pENTR insertion	CIBN-mCer-MP Fw	GCTGCTCCATTTACAATGAATGGAGCTATAGGAG
	CIBN-mCer-MP Rv	AGCGCGTCCACCTTTTCAATGGGGCAGGACG
	CRY2-V_H_H Fw	GCTGCTCCATTTACAATGGACAAAAAGACCATC
	CRY2-V_H_H Rv	AGCGCGTCCACCTTTTTAGCTGGAGACGGTGAC
	aPKC Fw	GCTGCTCCATTTACAATGCAGAAAATGCCCTC
	aPKC/aPKCΔN Rv	AGCGCGTCCACCTTTGACGCAATCCTCCAGAG
	aPKC ΔN Fw	GCTGCTCCATTTACAATGAAGCTGTTGGTGCACAAG
	Par6 Fw	GCTGCTCCATTTACAATGTCGAAGAACAAGATAAACAC
	Par6 Rv	AGCGCGTCCACCTTTCAAATGCAGCACTCCATC
	pENTR Fw	AAAGGTGGACGCGCTGACCCAGCTTTCTT
	pENTR Rv	TGTAAATGGAGCAGCCGCGGAGC
mutagenesis	CRY2olig E490G Fw	GATCTCTCGCACTCGGGGCGCCCAGATTATG
	CRY2olig E490G Rv	CGAGTGCGAGAGATCGCTTTAG
	Gateway AfeI Fw	GAGTCAGTGAGCGAGCACGTGGAAGAGCGCTCAATACGC
	Gateway AfeI Rv	CTCGCTCACTGACTCGCTGCGCTC
CPEC	Gateway CIBN Fw	TCCCGTTTGCGGCATTTTGCCTTC
	Gateway CIBN Rv	GGCGCTCAATAAGGGCGACACGGA
	Gateway CRY2 Fw	CCCTTATTGAGCGCCCAATACGCAAAC
	Gateway CRY2 Rv	AATGCCGCAAACGGGATCCAGACATGATAAG

CPEC: Circular Polymerase Extension Cloning.

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
