# Peer review of "Light-Induced Protein Clustering for Optogenetic Interference and Protein Interaction Analysis in Drosophila S2 Cells"

_biomolecules, 2019, doi:10.3390/biom9020061_

Reviewer 1 Report

Osswald's manuscript provided a detailed information about how an optogenetic method named as GFP-LARIAT can be applied to Drosophila S2 cell cultured system, in order to spatiotemporally photo-activate to do the clustering-trap of any GFP-tagged protein. The parts about how the method is built up and maximaized are quite informative to the researchers who will be interested in applying this technique in S2 cells. In addition, authors also showed other application of this optogenetic tool to study the protein-protein interaction. This might extend new field of studying protein-protein interaction and complexes in a spatio-tempral manner. In a global view, I strongly recommend the publication of this beautiful work.

To be perfect, I recommend two minor modification:

1, In the first paper of LARIAT (Nat Methods), Won Do Heo's lab found that reversible time period is around 10 min. However, this manuscript showed the complete recovery back to diffusive status needs 30 min. Could the authors clarify this difference? It might be due to laser power, or the amount of LARIAT expression level, or the amount of GFP-tagged protein?

2, In all previous papers about GFP-LARIAT (including mammalian cells and Drosophila in vivo), there is no any data or discussion about whether quick clustering trap of GFP-tagged proteins might trigger some potentially confounding cell responses. What is the fate of these aggregates after formation? Do they affect cell function or survival? If authors can do a little bit more tests, this will strongly improve the application of this technique in different projects related to S2 cells. At least, discussion about this potential side-effect is highly recommended.

Author Response

The reply is provided as the PDF file - Osswaldetal_reviewer1

Reviewer 2 Report

In this manuscript, Osswald and coauthors reported the adaptation of a Cry2-CIB1 based optogenetic system LARIAT in Drosophila S2 cells. The authors first constructed and compared various LARIAT modules for Drosophila S2 cell expression. They then demonstrated optogenetic interference of two proteins Mps1 and Lgl using the LARIAT system. In addition, a protein-protein interaction validation experiments were performed based on LARIAT induced co-clustering. Overall, this work represents a reasonable expansion of the LARIAT system in a model cell line. The manuscript is clear and concise, the experiments are technically sound. The only concern I have is that, although the authors successfully demonstrated localization of Mps1 and Lgl were disrupted by light using LARIAT system, it remains unclear to me that whether the actual function of Mps1 or Lgl was also inhibited by such dis-localization. As GFP-tagged Mps1 and Lgl were over-expressed in these experiments, it seems very likely the LARIAT system only traps a small portion of the targeted protein (i.e. Mps1 or Lgl) optogenetically, while the remaining protein can still be functioning. Functional assays of Mps1 and Lgl in live S2 cells should be included in order to exam the effectiveness of LARIAT protein inhibition within the experimental conditions described in this manuscript. Once this concern of mine has been addressed, I would recommend its publication in Biomolecules.

Author Response

We thank all reviewers for their positive evaluation and for the constructive remarks as we feel that by addressing the majority of their comments we have markedly improved our manuscript. As part of the revision, we have included new data and made important changes to the manuscript that we describe in the detailed answer to the specific points raised by each reviewer. In particular, we would like to draw your attention to: a) further analysis of parameters that control the kinetics of LARIAT protein clustering in Drosophila S2 cells (reviewer 1, Figure A1) b) the observation that LARIAT-mediated protein clusters do not cause apparent defects in mitotic progression and mitotic spindle organization (related to comments of reviewer 1 and reviewer 3, Figure 4), and c) the introduction of functional analysis of Mps1 protein inactivation by optogenetic trapping (related to comments of reviewer 2 and reviewer 3, Figure 5C,D).

Reviewer 2

In this manuscript, Osswald and coauthors reported the adaptation of a Cry2-CIB1 based optogenetic system LARIAT in Drosophila S2 cells. The authors first constructed and compared various LARIAT modules for Drosophila S2 cell expression. They then demonstrated optogenetic interference of two proteins Mps1 and Lgl using the LARIAT system. In addition, a protein-protein interaction validation experiments were performed based on LARIAT induced co-clustering. Overall, this work represents a reasonable expansion of the LARIAT system in a model cell line. The manuscript is clear and concise, the experiments are technically sound. The only concern I have is that, although the authors successfully demonstrated localization of Mps1 and Lgl were disrupted by light using LARIAT system, it remains unclear to me that whether the actual function of Mps1 or Lgl was also inhibited by such dis-localization. As GFP-tagged Mps1 and Lgl were over-expressed in these experiments, it seems very likely the LARIAT system only traps a small portion of the targeted protein (i.e. Mps1 or Lgl) optogenetically, while the remaining protein can still be functioning. Functional assays of Mps1 and Lgl in live S2 cells should be included in order to exam the effectiveness of LARIAT protein inhibition within the experimental conditions described in this manuscript. Once this concern of mine has been addressed, I would recommend its publication in Biomolecules.

This is an excellent suggestion as functional analysis of protein inactivation were missing in the previous version of the manuscript. Although S2 cells have been widely used to dissect mechanisms controlling Lgl subcellular localization (Betschinger et al., 2003; Carvalho et al., 2015; Graybill and Prehoda, 2014), its role in asymmetric stem cell division or mitotic spindle orientation in epithelia cannot be properly recapitulated in this system, making S2 cell culture inappropriate to analyze the functional outcome of its inactivation during cell division. Nevertheless, we would like to note that Lgl function is linked to its subcellular localization since its relevant function is at the cortex. So, we would expect that removal of Lgl from the cortex would interfere with protein function. 

More importantly, functional analysis of Mps1 is particularly relevant in S2 cell culture, which has been used to show that Mps1 overexpression enhances spindle assembly checkpoint signaling, blocking cells in mitosis (Conde et al., 2013). We have therefore examined if LARIAT-mediated clustering of Mps1 would be sufficient to reduce the duration of the mitotic arrest. The data now included in Figure 5, indicates that clustering of Mps1 indeed inhibits Mps1 activity as it reduces the mitotic timing of cells with high Mps1 overexpression. We would like to note that we agree with the reviewer statement that LARIAT may only partially trap and inactivate the protein of interest, if LARIAT is not used in excess to its target (as referred in the manuscript – “ We found that efficient removal of Lgl from the cortex requires high levels of RFP-CRY2 relative to Lgl-GFP (Figure 6B, B’, C), which highlights the importance of expressing LARIAT modules in excess…”). However, this new data indicates that if we ensure good expression levels of LARIAT components, we can effectively trap and inactivate proteins, even for functional analysis under overexpression conditions.

REFERENCES 

- Betschinger, J., Mechtler, K., and Knoblich, J.A. (2003). The Par complex directs asymmetric cell division by phosphorylating the cytoskeletal protein Lgl. Nature 422, 326-330. 

- Carvalho, C.A., Moreira, S., Ventura, G., Sunkel, C.E., and Morais-de-Sa, E. (2015). Aurora A triggers Lgl cortical release during symmetric division to control planar spindle orientation. Curr Biol 25, 53-60. 

- Conde, C., Osswald, M., Barbosa, J., Moutinho-Santos, T., Pinheiro, D., Guimaraes, S., Matos, I., Maiato, H., and Sunkel, C.E. (2013). Drosophila Polo regulates the spindle assembly checkpoint through Mps1-dependent BubR1 phosphorylation. EMBO J 32, 1761-1777. 

- Graybill, C., and Prehoda, K.E. (2014). Ordered multisite phosphorylation of lethal giant larvae by atypical protein kinase C. Biochemistry 53, 4931-4937.

Reviewer 3 Report

In this paper, Osswald, et al, adapted the Light activated reversible inhibition by assembled trap (LARIAT) from mammalian cells to Drosophila S2 cells for optogenetic perturbation of protein functions and detection of protein-protein interactions. They made multiple modifications on LARIAT for more convenient and effective applications in S2 cells: 1) they used a heat shock promoter to drive the expression of the LARIAT component proteins, which obviates the need to limit light exposure during cell culture; 2) they removed the mCerulean tag from the CIB1 component of LARIAT, which improved the tendency of LARIAT to form clusters in S2 cells; 3) they discovered LARIAT has different cluster dissociation kinetics when using different CRY2 variants, which could provide different temporal control over the optogenetic perturbation of protein functions; 4) finally, they used LARIAT to detect protein-protein interactions in S2 cells by examining the co-localization of the two proteins of interest in the clusters. Osswald, et al, also showed that LARIAT can be applied in S2 cells to effectively sequester proteins Mps1 and Lg1. Overall, I think this work is interesting.

I suggest the following modifications to improve the manuscript:

1.    Rewrite the introduction, the current introduction does not justify the work. 

2.    In figure 4, labeling RFP-CRY2/mCherry-Tub is confusing, it is not clear whether the red fluorescence comes from RFP-CRY2 or mCherry-Tub. Different colors should be used for Tubulin marker and CYR2. I suggestion changing RFP to BFP or mCherry to BFP, or simply use anti-tubulin immunostain.

3.    For figure 4, I suggest tracking the mitosis for a longer time to observe whether mitosis was perturbed due to optogenetic trapping. It would be interesting to see what would happen when light was applied at different stages of the mitosis. If a different color of tubulin was used, maybe we could already observe the perturbations. 

4.    In Figure 5A, where did the protein Lg1-GFP go at time point of 20 min? The current images shown in the manuscript did not seem to indicate that Lg1-GFP was removed from the cortex. It appears Lg1-GFP was still localized in the cortex but they form clusters. It would also be interesting to see whether sequestering of Lg1-GFP in the clusters will cause any malfunction?

5.    For Figure 6, it is interesting to use LARIAT to detect protein-protein interactions. This provides one more approach to validate protein-protein interactions (PPI) in single cell resolution! I suggest showing FLAG immunostain in the images as well. It is convenient to use existing GFP-aPKC construct. However, the expression level of GFP-aPKC and Par6-RFP will affect the outcome greatly. For a more robust assay to detect PPI, I suggest performing the same experiments with direct fusion of CRY2 to aPKC.

Author Response

The point-by-point response to reviewer 3 is submitted as a PDF file: Osswaldetal_reviewer3

Round  2

Reviewer 2 Report

The authors have done a great job of addressing my comments from the previous review of this manuscript. I recommend that the revised manuscript be suitable for publication.